# Sepsis Prediction by Using a Hybrid Metaheuristic Algorithm: A Novel Approach for Optimizing Deep Neural Networks

**DOI:** 10.3390/diagnostics13122023

**Published:** 2023-06-10

**Authors:** Umut Kaya, Atınç Yılmaz, Sinan Aşar

**Affiliations:** 1Faculty of Engineering and Architecture, Department of Software Engineering, İstanbul Beykent University, Istanbul 34398, Turkey; 2Faculty of Engineering and Architecture, Department of Computer Engineering, İstanbul Beykent University, Istanbul 34398, Turkey; 3Intensive Care Unit, Bakirkoy Dr. Sadi Konuk Training and Research Hospital, Istanbul 34147, Turkey

**Keywords:** artificial intelligence, sepsis, deep neural network, diagnosis

## Abstract

The early diagnosis of sepsis reduces the risk of the patient’s death. Gradient-based algorithms are applied to the neural network models used in the estimation of sepsis in the literature. However, these algorithms become stuck at the local minimum in solution space. In recent years, swarm intelligence and an evolutionary approach have shown proper results. In this study, a novel hybrid metaheuristic algorithm was proposed for optimization with regard to the weights of the deep neural network and applied for the early diagnosis of sepsis. The proposed algorithm aims to reach the global minimum with a local search strategy capable of exploring and exploiting particles in Particle Swarm Optimization (PSO) and using the mental search operator of the Human Mental Search algorithm (HMS). The benchmark functions utilized to compare the performance of HMS, PSO, and HMS-PSO revealed that the proposed approach is more reliable, durable, and adjustable than other applied algorithms. HMS-PSO is integrated with a deep neural network (HMS-PSO-DNN). The study focused on predicting sepsis with HMS-PSO-DNN, utilizing a dataset of 640 patients aged 18 to 60. The HMS-PSO-DNN model gave a better mean squared error (MSE) result than other algorithms in terms of accuracy, robustness, and performance. We obtained the MSE value of 0.22 with 30 independent runs.

## 1. Introduction

Sepsis is a serious disease that requires serious treatment and a long intensive care process, and it can be fatal if not diagnosed in time. Given the current epidemic, the limited capacity of critical care units might be a major issue for both patients and healthcare professionals. Antibiotic treatment for patients should be undertaken as soon as possible. Several studies in the literature use neural networks to predict sepsis [1,2,3,4,5,6,7,8,9,10,11,12,13,14,15]. In recent studies, sepsis and sepsis mortality have been predicted using a multilayer perceptron, neural networks with deep learning, Long Short-Term Memory (LSTM), and Recurrent Neural Networks (RNN). These models employ feedforward and back-propagation learning algorithms. Feedforward neural networks with a back-propagation algorithm that works with a fixed learning rate in the background can face many problems due to the hidden layer structure and the number of neurons. These problems include overlearning, undertraining, and not reaching the global minimum solution. Gradient-based algorithms are used in these models to adjust weights and back-propagate the error. Gradient-based methods initialize from a point of solution space and gradually move to the global minimum but sometimes get stuck at the local minimum. On the other hand, metaheuristic algorithms provide a different operation to escape from the local minimum to obtain the best chance.

The work goals to propose a model to solve the global minimum and other problems. Heuristic and metaheuristic algorithms can be used instead of gradient-based methods in solving real-world problems. Metaheuristic algorithms are population-based and start from multiple points and then cooperate. These algorithms have the potential to provide the closest result or global minimum to the solution of the problem.

Some studies refer to swarm intelligence, which can be used in the different artificial intelligence fields, the training of artificial neural networks, and machine learning [16,17,18,19]. Many hybrid algorithms have been proposed and analyzed; however, they are not capable of tuning the parameters of neural networks into different research. 

A standard algorithm version was chosen that includes the ability to modify the weights of a new type of neural network. Specific issues are dynamic and covered by other variants [20,21]. The multi-swarm method was created for dynamic problems, not for static problems, which means the parameters are considered vectors/weights to find minimum global optimization changes at the same time. In this study, the parameters are constant at each iteration and provided a space solution for global minimum optimization.

To predict sepsis for patients, the study is focused on the integration of a metaheuristic algorithm with a deep neural network (DNN). In this study, Particle Swarm Optimization (PSO) and Human Mental Search (HMS) are combined and used to tune the parameters of the DNN. Then, the proposed model (HMS-PSO-DNN) is applied to the sepsis dataset. Following that, a comparison of metaheuristic and gradient-based algorithms used in neural network optimization is mentioned in the literature.

Barkat et al. suggested the chicken flock optimization method. The suggested approach is compared to back-propagation neural networks, artificial bee colony back-propagation, and artificial neural networks using a genetic algorithm. They claim that their suggested method surpasses existing algorithms in accuracy and mean square error [22].

To improve the efficiency of neural network training in supervised learning, Ahmad and Mansour employed the cuckoo search algorithm, one of the metaheuristic methods for addressing optimization issues, in a multilayer feed-forward artificial neural network. They tested the network’s accuracy by solving classification problems with this trained network. They compared their findings to the Particle Swarm Optimization (PSO) and Guaranteed Convergence Particle Swarm Optimization (GCSPO) algorithms. They claimed that the suggested strategy outperformed the others on four separate categorization problems [23].

Irmak and Gülcü optimized the weight and bias settings when training an artificial neural network using the butterfly optimization method established by studying the behavior of butterflies. They compared their suggested model to the bat algorithm artificial neural network (ANN) model, the State of Matter Search algorithm model, and the back-propagation ANN model. According to the findings of the learning models they used for the XOR, balloon, breast cancer, and heart datasets, the model they presented outperformed the others [24]. Shi and Li assessed the performance of residential buildings using neural networks designed for ant colonies [25]. Sivagaminathan and Ramakrishnan presented a hybrid approach to trait subset selection based on neural networks and ant colony optimization [26]. For training feed-forward neural networks, Socha and Blum employed the ant colony optimization algorithm [27].

Dorigo et al. employed ant colony optimization to solve various optimization problems. It was created by [28]. By integrating PSO based on Newton’s laws of motion with accelerated central particle gravity optimization, Beheshti and Shamsuddin created the centripetal accelerated PSO approach. They claim that this approach, dubbed “extended PSO”, improves the ANN’s convergence and learning speed. They employed the proposed approach in conjunction with an ANN to diagnose diseases [29]. Mirjalili trained the multilayer perceptron using the gray wolf algorithm [30].

The firefly method was proposed by Brajevic and Tuba for training a feed-forward neural network [31]. For training the back-propagation neural network, Nandy et al. introduced the nature-inspired firefly algorithm [32]. Kovalski and Lukasik trained ANN using the Krill Swarm algorithm [33]. Devikanniga and Raj suggested an ANN model for osteoporosis classification based on monarch butterfly optimization [34]. Jaddie et al. optimized the neural network using a modified bat-inspired method [35]. Yaghini et al. established a hybrid back-propagation method with momentum-based anti-based PSO by integrating the capabilities of metaheuristic and greedy gradient-based algorithms. They claimed that utilizing the time-varying parameter increases the traditional PSO’s search capabilities and that the limitation factor assures particle convergence [36]. To solve the time series problem, Alweshah employed the firefly algorithm and ANN [37].

Karaboğa and Öztürk employed the Artificial Bee Colony Algorithm to train an ANN used to classify patterns [38]. Leung et al. suggested a sophisticated evolutionary method for fine-tuning the topology and parameters of the neural network [39]. To train the neural network, Yang and Kao created a powerful evolutionary algorithm [40]. Mirjalili suggested a hybrid approach for ANN training that combines gravitational search with PSO [41]. Donate et al. suggested a sophisticated ANN model that makes use of the genetic algorithm, differential evolution, and distribution estimation algorithm for estimating time series [42].

Da and Xiurun proposed an ANN based on PSO created using the simulated annealing approach [43]. To train the feed-forward ANN model, they created e-learning materials. Khan and Sahai compared the genetic algorithm, the PSO method, and the back-propagation algorithm to other algorithms [44]. Including Parsian by using the gray wolf optimization technique to optimize the ANN, they created a hybrid neural network model for melanoma diagnosis. They claim that their strategy improved the multilayer perceptron’s performance [45].

Yelghi et al. suggested a modified version of the firefly technique utilizing metaheuristics. The Adaptive Neural Fuzzy Inference System (ANFIS) adjusts the parameters of the clustering algorithm in an attempt to address difficult problems such as neural networks [46,47,48,49,50]. They have attempted to use optimization techniques to address complicated issues in the fields of finance and software engineering [51,52,53]. Therefore, resolving these issues will open new doors for the implementation of solutions to other genuine issues. This work’s contributions and novelty are as follows:The hybrid metaheuristic algorithm is proposed.Overlearning, insufficient learning, and memorization are attempted to be avoided by focusing on solving the local minimum problem encountered by gradient-based algorithms.A novel deep neural network design with optimum weights is proposed by the hybrid metaheuristic algorithm.The suggested model and topology are aimed to reduce the risk of sepsis in intensive care units.

The rest of this study has been arranged as follows: Section 2 presents the topology of the proposed algorithm, which is integrated with the DNN. Section 3 presents the proposed optimization algorithm. Section 4 mentions the experimental studies. These applications consist of the demonstration of the convergence ability of the proposed algorithm using benchmark functions, the experiments for the proposed algorithm with benchmark datasets, and the experiments for the proposed algorithm with the sepsis dataset. Section 5 describes the experimental results. These results are the convergence results of the proposed algorithm, the test results of the proposed algorithm integrated with the DNN for benchmark datasets, and experimental results for the sepsis dataset. Section 6 is the conclusion, which presents the general findings of the whole paper.

## 2. The Topology of the Proposed Algorithm Integrated with the DNN

Several deep learning architectures have been mentioned in the literature. Deep Convolutional Neural Networks, which are among these models, are used in image and sound recognition, image processing, and natural language processing, whereas LSTMs and variants of it are utilized in speech recognition, text and signal processing, and drug discovery. These models are more expensive to run, impractical, and unsuitable in circumstances such as data preparation and feature extraction than the deep neural network architecture used for sepsis prediction in the sepsis dataset, which solely comprises numerical data. Deep neural networks can perform the solution of complex and nonlinear real problems by forming relationships between the data.

The HMS-PSO algorithm is utilized to optimize the weights and biases (parameters) of neural network structures which are integrated called HMS-PSO-DNN. Figure 1 illustrates a flowchart of HMS-PSO-DNN. The model was trained by using the sigmoid activation function per neuron and applied to the benchmark dataset and real dataset, which will be discussed in the next section. All evaluations are based on the accuracy mean square error (*MSE*) and therefore investigated in this study. The formula of MSE is as follows. The mean squared error is calculated by Equation (1) [18].
(1)$$MSE=\frac{1}{U}\sum_{U=1}^{U}{\left(y_{u}-y_{u}^{ı}\right)}^{2}$$

The framework of HMS-PSO-DNN is indicated in Figure 2. As indicated earlier, the parameters of each layer should be tuned. All parameters convert to vectors or particles and they can be defined as one solution in metaheuristics. As we know, the metaheuristic algorithms based on the population size show the different solutions per iteration and optimize the solutions according to the paradigm of the algorithm. The number of adjustable parameters is calculated by Equation (2) [19].
(2)$$NN=\left(N_{i}+1\right)x\left|h_{1}\right|+\left(h_{1}+1\right)x\left|h_{2}\right|+\left(h_{2}+1\right)x\left|h_{3}\right|+\left(h_{3}+1\right)xN_{0}$$

To tune the hyperparameters, the model was trained by using the sigmoid activation function in neurons and applied to the dataset with 1200 instances and runs with 500 iterations. Based on the accuracy mean square error (MSE) computed and investigated and based on the experimental stochastic strategy, the “24-18-9-3-2” model was selected as the best topology structure. Table 1, below, shows this. Here, AUC is the area under the curve and MSE is the mean squared error.

## 3. The Proposed Optimization Algorithm

In this part, the Particle Swarm Optimization (PSO) and Human Mental Search (HMS) population-based metaheuristic algorithms were utilized together and called the HMS-PSO algorithm. The proposed HMS-PSO algorithm employs the HMS mental search operator, which imitates human behavior and includes the exploration and exploitation techniques of PSO. Depending on the scale of the problem and the optimization challenge, the classic PSO algorithm finds the minimum by a combination of exploration and exploitation operators related to the social coefficient, wherein exploitation refers to the cognitive coefficient in PSO. The advantage of HMS is a mental search that can be defined as a new extended exploitation. By considering the advantages of these algorithms, we provided a new hybrid algorithm. The suggested HMS-PSO algorithm’s pseudo code is displayed below [54].

In PSO, Equation (1) is used to calculate the updating particles, where *v* is the particle’s velocity, *w* is the inertia weight, *c*_1_ and *c*_2_ are the cognitive and social coefficients, and *r*_1_ and *r*_2_ are random vectors that regulate the stochastic impact of the cognitive and social components on the particle. *G_best_* is the global minimum value for the global best solution, and *P_best_* is the minimum value of the local best solution for each particle.
(3)$$v=w\;*\;v+c_{1}\;*\;r_{1}\;*\;\left(P_{best}-x\right)+c_{2}x\;*\;r_{1}\;*\;(G_{best}-x)$$

In Equation (3), *w* represents inertia weight, while *c*_1_ and *c*_2_ and *r*_1_ and *r*_2_ are randomly selected values. A high coefficient of inertia can make a particle travel faster, but a high coefficient of inertia can also make a particle drift away from the swarm. The particle may become trapped in its local optimal solution as a result of the cognitive coefficient selection. The social coefficient that is chosen may cause it to imitate the best global solution without discovering any alternative or superior global solutions.

The HMS-PSO method aids the particle by using the mental search operator to discover the optimal solution. The normal distribution *d* is a form that describes the random data. The random number is mc, which produces the conventional normal distribution in Equation (4). The layout is symmetrical because the majority of data hover close to the mean.
(4)$$d=\frac{1}{\sqrt{2\pi e^{\frac{{-mc}^{2}}{2}}}}$$

According to the HMS method in Equation (5), *Beta* is a random uniform distribution with a range of 0.3 and 1.99 [55].
(5)$$sigma=sin(\frac{\pi Beta}{2})$$

*S* denotes the number of successive values generated for each idea in the mental search. In this stage, a proposal based on the Levy flight, which is a random integer between the higher and lower bounds, determines a new solution. Levy flight is a unique variety of random walks in which step size is determined by the Levy distribution. The next location in a random walk is determined only by the present position in the solution space. The Levy flight involves many quick trips and numerous small steps. By way of explanation, the Levy flight simultaneously enhances the quality of exploration and exploitation.

Levy flight is more effective than Brownian motion for navigating ambiguous regions, which is an important distinction to make [55]. *S* is the Levy formula, which is shown in Equation (6), where *u* and *v* are random normal distribution numbers between 0 and 1, *i* is the number of particles, *x^i^* is the random uniform initial location of each particle, and 0.01 is the Levy distribution coefficient number [55].
(6)S=(d ∗ 0.01 ∗ u ∗ sigmav1Beta ∗ xi)

To obtain the best solution, the mental search operator is performed in a diverse manner along with the Levy flight distribution. According to the outcome of the mental search, the particle’s current position is *L_best_*, the local best position is *P_best_*, and *S* is the new position in Equation (7). While running the algorithm, the fitness value of positions is compared and the local best position is saved.
(7)$$if\;L_{best}<S\;then\;P_{best}=L_{best},\;if\;S<L_{best}\;then\;P_{best}=S$$

The distribution scale factor is represented by α, whereas Beta represents the distribution index, which ranges from 0.3 to 1.99. The suggested pseudo code of the HMS-PSO algorithm is presented below in Algorithm 1 [54].
**Algorithm 1.** The proposed algorithm pseudo code.Initialize particle populationfor t=1:maximum generation  Initialize global and local best particles pb and pg  for i=1:population size   vit+1 = wt ∗ vit+c1 ∗ r1 ∗ pb−xit+c2 ∗ r2 ∗ pg−xit     if vit+1 > vmax then vit+1=vmax     else ifvit+1 < vmin then vit+1=vmin     end     xit+1=xit+vit+1;     if xit+1 > xmax then xit+1=xmax     else if xit+1 < xmin then xit+1=xmin     end   endInitialize the mental search for each particlefor i = 1:population size  S=d ∗ 0.01 ∗ u ∗ sigma(v1Beta) ∗ xi  xs=S   if fxit<fpbt then pbt=xit   else if fxst<fpbt then pbt=xst   end end fpgt<minifpbt wt=tmax−ttmaxwmax−wmin+wminend

## 4. Experimental Studies

Proposed method’s convergence to the minimum and maximum values with using benchmark functions is demonstrated in this section. Because metaheuristic algorithms cannot be proven, convergence criteria are used to show algorithms. The network design for performing the sepsis classification problem is established when the created hybrid metaheuristic algorithm is integrated with deep neural networks. The network architecture developed as a result of experimental research is used to predict sepsis. 

The proposed HMS-PSO algorithm and other algorithms, called PSO and HMS, were implemented using the Python programming language in the Jupyter Notebook environment installed on Windows 10. The processor was an Intel(R) Core(TM) i7-8750H CPU @ 2.20 GHz. The Python libraries used included NumPy, Math, Random, Matplotlib, Pandas, and Scipy. The iris, wine, and breast cancer benchmark datasets were obtained using the Sklearn package. Sklearn Preprocessing MinMaxScaler was used to standardize the data, Sklearn Simple Imputer was used to replace missing data, and Sklearn train-test-split was used to separate the dataset into training and test. The data were divided into two pieces: training 80% and testing 20%. One hot encoding was performed on the outputs to work on increasing the classification accuracy.

### 4.1. Demonstration of the Convergence of the Proposed Algorithm Using Benchmark Functions

The performance of the HMS-PSO algorithm was compared to PSO and HMS algorithms using benchmark test functions. According to the benchmark test functions, algorithms with 30 dimensions and 2 dimensions were each performed 30 times. The suggested HMS-PSO method outperforms the PSO and HMS algorithms, according to performance measurements.

Metaheuristic algorithms could not be demonstrated with mathematical proof due to being implanted based on the stochastic strategy. Instead, for surveying the effectiveness of the algorithms according to whether algorithms lead to convergence in the global best position or close to the global best position, the proposed and compared algorithms were applied to the benchmark functions. In this section, benchmark functions were utilized and defined with limitations and features, which are indicated in Table 2, and the remaining benchmark functions refer to [56,57,58].

The experiment was performed with 30 runs on 30 and 2 dimensions of functions. For all algorithms, we considered 50 population sizes and 100 iterations, equaling approximately 5000 algorithms to evaluate and provide the solutions. The most widely used parameter values in the literature were selected for convergence analysis. The determined parameters of the algorithms for the benchmark functions are presented in Table 3.

### 4.2. Experiments for the Proposed Algorithm with Benchmark Datasets

To investigate and acquire the proper topology of the structure, the model was trained by using the sigmoid activation function in neurons and applied to the benchmark dataset. The experimental topology for DNN based on the accuracy mean square error (MSE) was investigated with a different topology [26], which was applied to the benchmark dataset. The proposed model based on MSE was applied to different datasets and compared to other models. The most frequently used parameter values in the literature were selected for the analysis of DNN performance on benchmark datasets. The determined parameters of algorithms for the benchmark datasets are presented in Table 4.

### 4.3. Experiments for the Proposed Algorithm with Sepsis Dataset

After performing and investigating the benchmark dataset, the proposed model was implemented to the real dataset. The dataset included the vital records of 640 patients, who ranged in age from 18 to 60. All data were acquired under the direction of doctors in the intensive care unit. In the study, the patient’s gender and height were taken into consideration, along with the RR (respiratory rate), LYM (lymphocyte), LYM100 (lymphocyte ratio), WBC (white blood cell), neu (neutrophil), plt (thrombocyte), CRP (c-creative protein), Procalcitonin, HR (heart rate), spo2 (peripheral capillary oxygen saturation), Temp (body temperature), BPSYS (systolic blood pressure), bpDias (diastolic blood pressure), bpMean (mean blood pressure), Apache2_first (disease severity classification), Sofa_first (sequential organ failure assessment), and Apache2_Mort (disease mortality severity classification).

The deep neural network applied to the sepsis dataset was trained using the HMS-PSO, HMS, PSO, Gradient, Adadelta, and Stochastic Gradient Descent (SGD), RMSprop, and Adam optimization methods. The test dataset’s mean square error (MSE) and root mean square error (RMSE) values were obtained for comparison. The deep neural network was run 30 times independently and 2000 times with each algorithm. For back-propagation techniques, the learning rate was chosen by default. For each method, the sigmoid activation function was chosen. The determined topology for the DNN was applied to the sepsis dataset.

## 5. Discussion

### 5.1. Convergence Results of the Proposed Algorithm

The convergence analysis of the algorithms was performed using benchmark functions. The summarized details are compared in Table 5. The proposed algorithm, HMS, and PSO were applied to the benchmark functions. As can be seen in Table 5, HMS-PSO showed the best performance for convergence analysis.

The second column of the table shows the best algorithms for the related function, and based on the “No Free lunch” theory, there is no one algorithm best for optimization functions [59]. We can see that two or three algorithms show approximately the same results.

Based on the 30 runs for all algorithms, we observe the results of minimum value and maximum value.

The proposed HMS-PSO algorithm with the advantage of HMS, which is the extension exploitation, was combined with PSO. In the PSO algorithm, the exploration and exploitation operations sometimes become stuck at the local minimum and cannot cooperate and balance with each other. To solve the problem of PSO, we used the exploitation operation HMS, which provides a small step size in the deep solution space and a big step size in the hill of the solution space. The exploitation operation of HMS shows the random walk, which dynamically changes in the solution space, which is inspired by the Levy flight distribution.

The HMS-PSO algorithm converges in space solutions such as Rosenbrock, Rastrigin, Ackley, Griewank, Whitley, Schwefel_12, Schwefel_21, Hyper_Sphere, and McCormick benchmark functions and also runs on the remaining benchmark functions showing close values to global best. Based on the “No Free lunch” theory, this study is not expected to provide the best results of the whole metaheuristic algorithm. By considering the entire results of the experiment, we observe that the proposed algorithm shows suitable performance with different functions.

### 5.2. The Test Results of the Proposed Algorithm Integrated with DNN for Benchmark Datasets

The proposed algorithm was integrated with the DNN and the integrated model was implemented to determine the best topology with the sample dataset. In the experiment of the sample sepsis dataset, “24-18-9-3-2” was chosen as the best topology structure. In Table 6, the statistical analysis of all algorithms for benchmark datasets is given. The benchmark datasets are iris, breast cancer, and wine. Upon observation, the proposed model considering the best values indicates the best performance and robustness compared to the other models, according to MSE values.

Based on the 30 runs of all algorithms on the related dataset, the average (avg.), maximum (max.), minimum (min.), standard deviation (std.), and variance (var.) were computed.

### 5.3. Experimental Results for Sepsis Dataset

Statistical analyses based on MSE and RMSE for all algorithms run on the sepsis dataset are indicated in Table 7 and Table 8, respectively. The values are included to show the robustness and performance of the algorithms on the sepsis dataset. The proposed algorithm has the best MSE and RMSE values.

## 6. Conclusions

This research presents a new hybrid metaheuristic algorithm for the prediction of the sepsis dataset. In the work, we first tried to propose a hybrid metaheuristic algorithm to solve the global optimization and demonstrated its correctness with benchmark functions. Then, we integrated the proposed metaheuristic algorithm into a deep neural network.

In the experiment, all models were applied to the benchmark dataset to show reliability, robustness, and correctness. As a result, according to the statistical and other analysis metrics, the proposed model shows the optimal performance in comparison with the other applied models. After achieving a correct and robust model, we applied it to the sepsis dataset.

This data were obtained from Sadi Konuk Training and Research Hospital. The results indicate that a total of 216 of 640 individuals between the ages of 18 and 60 had sepsis, whereas the remaining 424 did not have sepsis.

Dataset parameters were chosen by medical specialists in the intensive care unit. The output values were also converted into two-dimensional vectors by applying one hot encoding to the Sklearn preprocessing library. These vectors consist of 0 s and 1 s.

The deep neural network applied to the sepsis dataset was trained using the HMS-PSO, HMS, PSO, Gradient, Adadelta, and Stochastic Gradient Descent (SGD), RMSprop, and Adam optimization methods. The test dataset’s mean square error (MSE) and root mean square error (RMSE) values were obtained for comparison. The deep neural network was run 30 times independently for each algorithm. The sigmoid was chosen for activation because it is more successful in binary classification. For back-propagation techniques, the learning rate was chosen by default. For each method, the sigmoid activation function was chosen. The determined topology for DNN based was applied to the sepsis dataset.

The statistical MSE and RMSE analysis of the proposed algorithm has shown robustness and the best performance on the sepsis dataset. The proposed algorithm succeeded in solving the classification problem of sepsis prediction. Since the sepsis dataset was unstable, it learned 0s better than 1s. A lower MSE value is associated with population size and a more appropriate selection of parameters. It is thought that the model can provide better results with a larger and more balanced dataset.

## Figures and Tables

**Figure 1 diagnostics-13-02023-f001:**
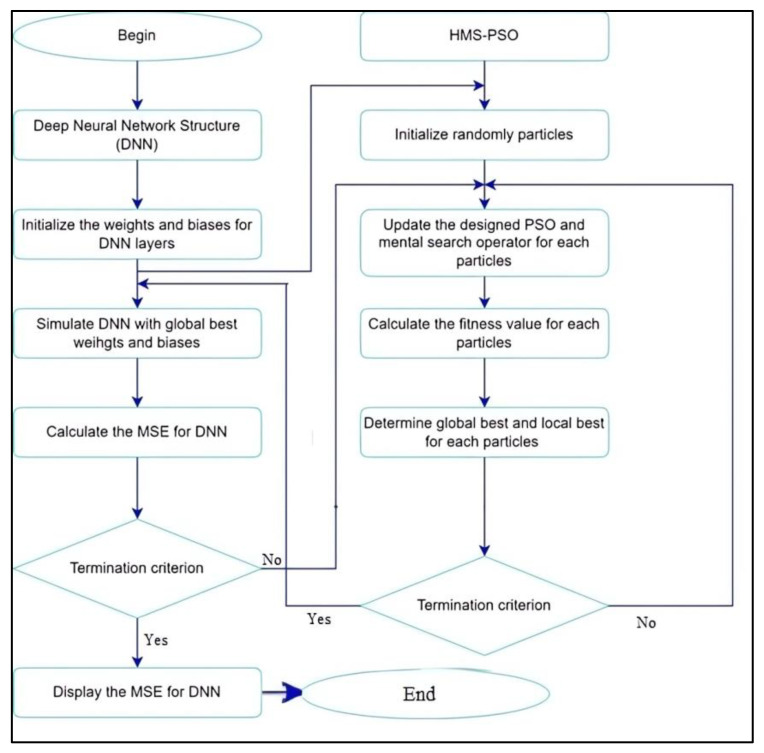
Flowchart of HMS-PSO-DNN.

**Figure 2 diagnostics-13-02023-f002:**
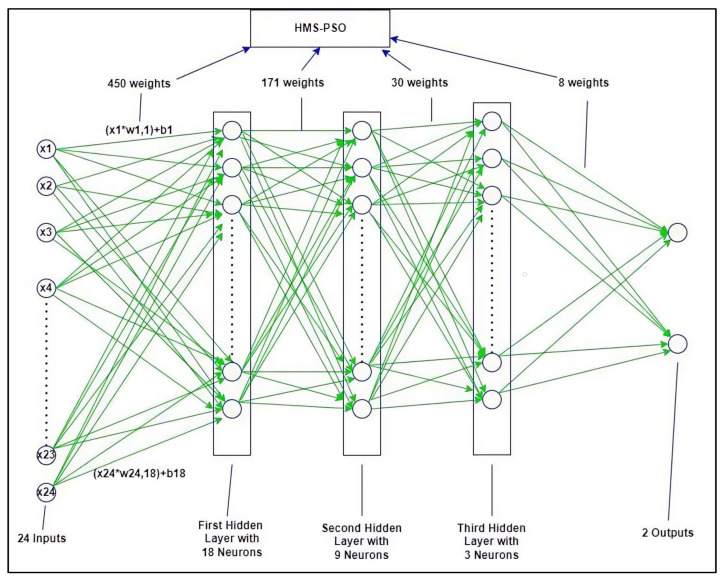
A sample topology of the HMS-PSO-DNN algorithm.

**Table 1 diagnostics-13-02023-t001:** Network models.

Number of Input Layer Neurons	Number of Hidden Layer Neurons	Output Layer Neurons	MSE	AUC
24	40-40-20-10-3	2	0.1791	0.5
24	20-40-9-6-3	2	0.1791	0.5
24	10-40-9-3	2	0.1791	0.5
24	10-20-30-5-2	2	0.1791	0.5
24	40-40-90-30	2	0.0791	0.83
24	50-15-18-3	2	0.1041	0.78
24	10-30-10-30	2	0.1041	0.76
24	70-35-30-7	2	0.1583	0.78
24	10-40-10-5	2	0.1375	0.71
24	30-25-10-3	2	0.0875	0.78
24	25-15-10-2	2	0.1416	0.65
24	21-10-5-3	2	0.175	0.51
24	20-20-30-50-50	2	0.0833	0.84
24	30-30-70-10	2	0.1791	0.5
24	3-15-9-2	2	0.125	0.796
24	21-12-9-3	2	0.1291	0.81
24	18-9-3	2	0.0791	0.897
24	18-12-9-3	2	0.075	0.8
24	18-12-6	2	0.1333	0.66
24	30-50-6-3	2	0.1791	0.5
24	15-18-12-3	2	0.1791	0.5
24	7-15-9-3	2	0.1416	0.74
24	18-12-6-3	2	0.1125	0.786
24	9-15-8-3	2	0.1041	0.73
24	12-18-9-3	2	0.1541	0.815
24	16-8-5-2	2	0.1791	0.5

**Table 2 diagnostics-13-02023-t002:** The benchmark functions with features.

Function	Function Definition	V no	Boundary	F min
Hyper Sphere	$F_{1}\left(x\right)=\sum_{i=1}^{n}x_{i}^{2}$	30	[−100, 100]	0
Shifted Schwefel’s Problem 1.2	$F_{2}\left(x\right)=\sum_{i=1}^{n}x_{i}^{2}\left|x_{i}\right|+\prod_{i=1}^{n}\left|x_{i}\right|$	30	[−10, 10]	0
Schwefel 1.2	$F_{3}\left(x\right)=\sum_{i=1}^{n}({\sum_{j=1}^{i}x_{j})}^{2}$	30	[−100, 100]	0
Schwefel 2.21	$F_{4}\left(x\right)=\max\left\{\left|x_{i}\right|,1\leq i\leq n\right\}$	30	[−100, 100]	0
Rosenbrock	$F_{5}\left(x\right)=\sum_{i=1}^{n-1}\left[{100\left(x_{i+1}-x_{i}^{2}\right)}^{2}+{\left(x_{i}-1\right)}^{2}\right]$	30	[−30, 30]	0
Step2	$F_{6}\left(x\right)=\sum_{i=1}^{n}{\left([x_{i}+0.5]\right)}^{2}$	30	[−100, 100]	0
Quartic	$F_{7}\left(x\right)=\sum_{i=1}^{n}ix_{i}^{4}+random\;[0,1]$	30	[−1.28, 1.28]	0
Schwefel	$F_{8}\left(x\right)=\sum_{i=1}^{n}{-x}_{i}sin(\sqrt{\left|x_{i}\right|})$	30	[−500, 500]	−418.9829 × 5
Rastrigin	$F_{9}\left(x\right)=\sum_{i=1}^{n}{[x}_{i}^{2}-10cos(2\pi x_{i})+10]$	30	[−5.12, 5.12]	0
Ackley	$F_{10}\left(x\right)=-20\exp\left(\sqrt[{-0.2}]{\frac{1}{n}\sum_{i=1}^{n}x_{i}^{2}}\right)-\exp\left(\frac{1}{n}\sum_{i=1}^{n}\cos\left(2\pi x_{i}\right)\right)\;\;\;\;\;\;\;+20+e$	30	[−32, 32]	0
Griewank	$F_{11}\left(x\right)=\frac{1}{4000}\sum_{i=1}^{n}x_{i}^{2}-\prod_{i=1}^{n}\cos\left(\frac{x_{i}}{i}\right)+1$	30	[−600, 600]	0
Six Hump	$F_{16}\left(x\right)={4x}_{1}^{2}-2.1x_{1}^{4}+\frac{1}{3}x_{1}^{6}+x_{1}x_{2}-4x_{2}^{2}+4x_{2}^{4}$	2	[−5, 5]	−1.0316
Branin	$F_{17}\left(x\right)={\left(x_{2}-\frac{5.1}{{4\pi }^{2}}x_{1}^{2}-6\right)}^{2}+10\left(1-\frac{1}{8\pi }\right){cosx}_{1}+10$	2	[−5, 5]	0.398
Gold Stein and Price	$F_{18}\left(x\right)=[1+{\left(x_{1}+x_{2}+1\right)}^{2}(19-14x_{1}+3x_{1}^{2}-14x_{2}+6x_{1}x_{2}\;\;\;\;\;\;\;+3x_{2}^{2})]\times [30+{\left(2x_{1}-3x_{2}\right)}^{2}\times (18-32x_{1}\;\;\;\;\;\;\;+12x_{1}^{2}+48x_{2}-36x_{1}x_{2}+27x_{2}^{2})]$	2	[−2, 2]	3

**Table 3 diagnostics-13-02023-t003:** Parameters of algorithms for benchmark functions.

Parameters	Definition	Value
W	Inertia values	0.729
C1	Local coefficient	1.49
C2	Global coefficient	1.49
ML	Mental lower bound	0.3
MH	Mental higher bound	1.99
UB	The upper bound of particles	10
LB	Lower bound of particles	2
N_var	Number of dimensions	30, 2
Cs	Cluster size	5

**Table 4 diagnostics-13-02023-t004:** Determined parameters of algorithms for benchmark datasets.

Parameters	Definition	Value
W	Inertia values	0.6
C1	Local coefficient	1.45
C2	Global coefficient	1.45
ML	Mental lower bound	0.3
MH	Mental higher bound	1.99
UB	The upper bound of particles	10
LB	Lower bound of particles	2
N_var	Number of dimensions for sepsis	640
Cs	Cluster size	5

**Table 5 diagnostics-13-02023-t005:** The summary of convergence performance for the algorithms implemented.

Function	The Best-Performing Algorithm	Min (Value)
Rosenbrock	HMS-PSO	0.000000005
Hyper_Sphere	HMS-PSO	0.04846808
Schwefel_12	HMS-PSO	0.35624187
Schwefel_21	HMS-PSO	1.49143852
Step2	HMS-PSO, PSO, HMS	0–0
Quartic	HMS-PSO, PSO, HMS	0–0
Schwefel	HMS	−4.00 × 10^197^
Rastrigin	HMS-PSO	11.06655678
Ackley	HMS-PSO	0.10581148
Griewank	HMS-PSO	0.58381321
Branin	HMS-PSO, PSO	0.39788736
Six_Hump_Camel	HMS-PSO, PSO	−1.03162845
Goldstein Price	HMS-PSO, PSO, HMS	3–3
Dejong	HMS-PSO, PSO	0–0
Schubert	HMS-PSO, PSO	−186.7309
Whitley	HMS-PSO	0.00000013
Matyas	HMS	4.99 × 10^−20^
Zakharov	HMS-PSO, PSO	0.05094591
McCormick	HMS-PSO	−149.2967921
Bohachevsky	HMS	6.64 × 10^−4^
Michalewicz	HMS-PSO, PSO	−1.91988

**Table 6 diagnostics-13-02023-t006:** The statistical analysis of all algorithms for benchmark datasets.

Dataset	Model	Mse (Avg.)	Mse (Min.)	Mse (Max.)	Mse (Std.)	Mse (Var.)
Iris	PSO-DNN	0.21066	1.50000	0.00000	0.28854	0.08325
	PROPOSED ALGORITHM	0.06440	0.00000	0.43330	0.09901	0.00980
	HMS-DNN	1.29167	1.29167	1.29167	0.00000	0.00000
Wine	PSO-DNN	0.41570	1.00000	0.00000	0.31860	0.10151
	PROPOSED ALGORITHM	0.16850	0.00000	0.94440	0.22495	0.05060
	HMS_DNN	1.37931	1.37931	1.37931	0.00000	0.00000
Breast Cancer	PSO-DNN	0.05850	0.01750	0.37720	0.06931	0.00480
	PROPOSED ALGORITHM	0.04150	0.01750	0.08770	0.01842	0.00033
	HMS-DNN	0.60440	0.60440	0.60440	0.00000	0.00000

**Table 7 diagnostics-13-02023-t007:** The statistical analysis based on MSE for all algorithms on sepsis.

Optimizer	MSE (Avg.)	MSE (Max.)	MSE (Min.)	MSE (Std.)	MSE (Var.)
HMS-PSO	0.25111	0.27995	0.22005	0.01567	0.00025
HMS	0.35807	0.35807	0.35807	0.00000	0.00000
PSO	0.34360	0.35807	0.27409	0.02535	0.00064
Grad	0.46931	0.64193	0.35742	0.13833	0.01914
Adadelta	0.39312	0.68294	0.31966	0.12399	0.01537
Sgd	0.33887	0.36003	0.31836	0.00966	0.00009
Rmsprop	0.27858	0.29753	0.25977	0.00913	0.00008
Adam	0.28676	0.35417	0.23177	0.03166	0.00100

**Table 8 diagnostics-13-02023-t008:** The statistical analysis based on RMSE for all algorithms on sepsis.

Optimizer	RMSE (Average)	RMSE (Max)	RMSE (Min)	RMSE (Standard Deviation)	RMSE (Variance)
HMS-PSO	0.50087	0.52910	0.46910	0.01567	0.00025
HMS	0.59839	0.59839	0.59839	0.00000	0.00000
PSO	0.58577	0.59839	0.52354	0.02224	0.00049
Grad	0.67813	0.80120	0.59785	0.09892	0.00979
Adadelta	0.62088	0.82640	0.56539	0.08883	0.00789
Sgd	0.58207	0.60002	0.56423	0.00829	0.00007
Rmsprop	0.52774	0.54546	0.50967	0.00863	0.00007
Adam	0.53475	0.59512	0.48143	0.02888	0.00083

## Data Availability

Not applicable.

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
