# Peer review of "Sepsis Prediction by Using a Hybrid Metaheuristic Algorithm: A Novel Approach for Optimizing Deep Neural Networks"

_diagnostics, 2023, doi:10.3390/diagnostics13122023_

Round 1

Reviewer 1 Report

A hybrid metaheuristic algorithm was proposed for the optimization with regards to the weights of the deep neural network and applied for the early diagnosis of Sepsis. The proposed algorithm aims to reach the global minimum with a local search strategy capable of exploring and exploiting particles in Particle Swarm Optimization (PSO) and using the mental search operator of the Human Mental Search algorithm (HMS).

HMS-PSO is integrated with deep neural networks (HMS-PSO-DNN). The study focused on predicting Sepsis with HMS-PSO-DNN utilizing a data set of 640 patients aged 18 to 60. The HMS-PSO-DNN model gave a better MSE result than other algorithms in terms of accuracy, robustness, and performance.

The structure and writing are good. Minor changes are required. Please consider the following comments to improve the paper.

1.      There are many PSO variants. Why do you use this version? Review for existing PSO variants is recommended to be added, e.g., A novel multi-swarm algorithm for optimization in dynamic environments based on particle swarm optimization, Comprehensive learning particle swarm optimization algorithm with local search for multimodal functions, etc.

2. Fig. 2 Flowchart of HMS-PSO-DNN should be modified. The outputs of judgements should be clear.  “Yes” and “No” should be out of the box and on the line. “End” cannot be found in the figure.

Author Response

Responds for Reviewer_1

The structure and writing are good. Minor changes are required. Please consider the following comments to improve the paper.

  1. There are many PSO variants. Why do you use this version? Review for existing PSO variants is recommended to be added, e.g., A novel multi-swarm algorithm for optimization in dynamic environments based on particle swarm optimization, Comprehensive learning particle swarm optimization algorithm with local search for multimodal functions, etc.
  2. 2 Flowchart of HMS-PSO-DNN should be modified. The outputs of judgements should be clear. “Yes” and “No” should be out of the box and on the line. “End” cannot be found in the figure.

Responds:

  1. This version was chosen standard algorithm and includes the ability to modify the weights of a new type of neural network. Specific issues are dynamic and covered by other variants [1-2]. The multi-swarm method was created for dynamic problems not for static problem, which means the parameters are considers as vector/weights to find minimum global optimization changed at the same times, in this study the parameters are constant at each iteration and provided a space solution for global minimum optimization. For better direction of reader and researcher we mentioned in this study.
  2. The mentioned Figure was modified and after edition it was Fig 1.

References:

[1]        Xuewen Xia, Ling Gui, Zhi-Hui Zhan, A multi-swarm particle swarm optimization algorithm based on dynamical topology and purposeful detecting, Applied Soft Computing, Volume 67, 2018, Pages 126-140,

[2]        H. Song, A. K. Qin, P. -W. Tsai and J. J. Liang, "Multitasking Multi-Swarm Optimization," 2019 IEEE Congress on Evolutionary Computation (CEC), Wellington, New Zealand, 2019, pp. 1937-1944, doi: 10.1109/CEC.2019.8790009.

Reviewer 2 Report

1. From abstract the dataset looks relatively small dataset of 640 patients, which may limit the generalizability of the results.

2. Introduction is adequate and provided the relevant aspects. Background elements is adequate

3. So first the proposed optimizer is discussed and then then topology of the algorithm. However, it should be opposite. First authors should discuss and provide the basic algorithm which is DNN. Further, author mentioned LSTMs are utilized in speech recognition, text and signal processing, and drug discovery. RNNs, which are generally employed to tackle unsupervised learning issues, is not a suitable method for supervised learning problems. Authors need to provide the foundation algorithm on which they have done the optimization. Also the particle swarm algorithm is part of genetic algorithms. It is not clear that author problem of research is relevant or not. Further, why not other techniques of GA? Authors mentioned that they have used neural network but writes only Deep Neural Network ? what is the specific model or algorithm used ? Need more information on layers too

4. Now in the experimental studies "HMS-PSO, PSO, and HMS algorithms are implemented using the Python programming language" but as per the previously mentioned method only HMS-PSO are mentioned. If this part of the study, then this will require a comparative study first on these and during experiments how these are worked. Further, is there any hyperparameter tunning being done ? as it is not clear that on which data authors are performing experimental studies. What is the source?

5.  authors mentioned "For all algorithms, we consider 50 population sizes and for iteration, the size is 100 which is about 500 function evaluation numbers function." explanation on using the specific iteration is needed and which algorithms ? only parameters are provided

6. Similarly, MSE cannot be the only reliable source. Need more evaluation metrics.

7. Now in 4.3 author mentioned the dataset but it has 640 records only. Very few for a deep learning model to be tested

8. Table 4 provided performance of more than 1 algorithm which was part of the methods. 

9. Discussion is inadequate. 

10. Now in 5.2 authors have more evaluation metrics But these are now based on PSO_DNN not HMS-PSO. Requires clarity

11. Table 6 and 7 have provided separate evaluation metrics for each HMS, PSO, Grad etc. But not the combined as authors have mentioned

12. This research presents a new hybrid metaheuristic algorithm for the prediction of Sepsis dataset. However, this is not the case. This research on combination of algorithms specially these are optimizers not the actual deep learning models or algorithms are not new. 

Author Response

Responds for Reviewer_2

Thank you for your valuable contribution to the manuscript. The corrections you specified have been completed.

  1. From abstract the dataset looks relatively small dataset of 640 patients, which may limit the generalizability of the results.
  2. Introduction is adequate and provided the relevant aspects. Background elements is adequate.
  3. So first the proposed optimizer is discussed and then topology of the algorithm. However, it should be opposite. First authors should discuss and provide the basic algorithm which is DNN. Further, author mentioned LSTMs are utilized in speech recognition, text and signal processing, and drug discovery. RNNs, which are generally employed to tackle unsupervised learning problems. Authors need to provide the foundation algorithm on which they have done the optimization. Also the particle swarm algorithm is part of genetic algorithms. It is not clear that author problem of research is relevant or not. Further, why not other techniques of GA? Authors mentioned that they have used neural network but writes only Deep Neural Network? What is the specific model or algorithm used? Need more information on layers too.

  1. Now in the experimental studies “HMS-PSO, PSO, and HMS algorithms are impelmented using the Python programming language” but as per the previously mentioned method only HMS-PSO are mentioned. If this part of the study, then this will require a comparative study first on these and during experiments how these are worked. Further, is there any hyperparameter tunning being done? As it is not clear that on which data authors are performing experimental studies. What is the source?
  2. Authors mentioned “For all algorithms, we consider 50 population sizes and for iteration, the size is 100 is which is about 500 function evaluation numbers function.” Explanation on using the specific iteration is needed and which algorithms? Only parameters are provided.
  3. Similarly, MSE can not be the only reliable source. Need more evaluation metrics.
  4. Now in 4.3 author mentioned the dataset but it has 640 records only. Very few for deep learning model to be tested.
  5. Table 4 provided performance of more than 1 algorithm which was part of the methods.
  6. Discussion is inadequate.
  7. Requires clarity.Now in 5.2 authors have more evaluation metrics. But these are now based on PSO_DNN not HMS-PSO.
  8. Table 6 and 7 have provided separate evaluation metrics for each HMS, PSO, Grad etc. But not the combined as authors have mentioned.
  9. This research presents a new hybrid metaheuristic algorithm for prediction of Sepsis dataset. However, this is not the case. This research on combination of algorithms specially these are optimizers not the actual deep learning models or algorithms are not new.

Response

  1. The dataset was large, But there is some preprocessing this dataset. Data over 2000 was available and consisted of 48 sections. However, there were patients aged 90 years and missing some data. The data set was shrunk since the age of 18-60 was taken as the criterion. However, 48 hours is not a valid time frame for Sepsis prediction. The important thing is to do a diagnosis as soon as possible. So he had 12 hours. In addition, the result of the 6-hour Sepsis forecast is also available. Normalization and data set connection operations were also performed on the data set. The reason of the above mentioned, and the advantage of DNN which is more suitable to overcome the more dimensions and overlapped dataset. Overall, we preprocessing operation and using the nonlinear classification of the Deep Neural Network.
  2. Thanks
  3. We edited and changed the section orders as review asks. We also edited the sentence of “Several deep learning architectures have been mentioned in the literature. Deep Convolutional Neural Networks, which are among these models, are used in image and sound recognition, image processing, and natural language processing, whereas LSTMs and variant of it are utilized in speech recognition, text and signal processing, and drug discovery.”

GA and PSO algorithm are metaheuristic algorithms, but they have difference in paradigm, GA is evolutionary computing based on the crossover, mutation and selection operator, but PSO is swarm intelligence and it is based on the local and global best position and depends on the last population based on the formula.

We added the more information as reviewer ask “to tune the hyper parameters the model was trained by using the sigmoid activation function in neurons and applied on the dataset with 1200 instances and with runs 500 iterations. Based on the accuracy mean square error (MSE) are computed and investigated, based on the experimental with stochastic strategy the “24-18-9-3-2” model was selected as the best topology structure. Table 1 below shows this. ”

  1. The base of PSO and HMS are mentioned formula 3 and 5 respectively. We combined and considered them and reformulated for our algorithm. Table 2 is added and description is detailed.
  2. The experiment has been performed with 30 runs on 30 and 2 dimensions of functions. For all algorithms, we considered 50 population sizes, iteration is 100 which is about approximately 5000 are evaluated and provide the solutions.
  3. MSE and RMSE are metrics and of course the proposed algorithm (HMS-PSO) is optimization algorithm but the importance in this study the classic method optimizations are used in DNN does not indicate the tune parameters of the DNN. For improving the tune the parameters of the DNN, we proposed the optimization algorithm HMS-PSO and also we compared with Adam, Adadelta, SGD ve RMS-PROP,HMS and PSO which those are combined with DNN and tune the parameters.
  4. We also used 3 data sets in Table 5 (iris, wine and breast cancer).
  5. The second column of the table is best algorithms on the related function and based on the no free lunch theorem there is not one algorithm best for optimization function. We can see two or three algorithm is approximately the same results. And more information to the discussion.
  6. The second column of the table is best algorithms on the related function and based on the no free lunch theorem there is not one algorithm best for optimization function. We can see two or three algorithm is approximately the same results. And more information to the discussion.
  7. As mentioned before and edited we firstly developed the optimization algorithm and tried to demonstrate the our algorithm that sway we need to comparison also after combining the proposed algorithm to the DNN we should to try more demonstrate and apply on the real dataset.
  8. As mentioned before and edited we firstly developed the optimization algorithm and tried to demonstrate the our algorithm that sway we need to comparison also after combining the proposed algorithm to the DNN we should to try more demonstrate and apply on the real dataset.
  9. As mentioned before and edited we firstly developed the optimization algorithm and tried to demonstrate the our algorithm that sway we need to comparison also after combining the proposed algorithm to the DNN we should to try more demonstrate and apply on the real dataset.

We know the work include some misunderstand of the states, table and section for the reader. By considering the suggestion of the author tried to be edited. Thank you very much.

Author Response

Responds for Reviewer_3

Thank you for your valuable contribution to the manuscript. The corrections you specified have been completed.

  1. The authors proposed a hybrid metaheuristic algorithm to solve the global optimization and demonstrated its correctness with various benchmark functions. The authors have then integrated the proposed metaheuristic algorithm with a deep neural network. The authors claimed that the proposed model gave better results than other algorithms. Although the authors formulate the problem well but I miss their proposition’s advantages in comparison to other works they already mentioned in the introduction. It is a bit confusing as the authors in section 5.2. Mentioned that the proposed algorithm was integrated with DNN, etc. Then table 5 shows the results of the proposed algorithm but not the one integrated with DNN!, How the best topology structure is chosen? What is the choice of the three machine learning algorithms? How the MSE and RMSE results (as well in other tables) are justified? In addition, It is more readable if the proposed system and the good results are highlighted in the tables. The results in tables from 4-7 need clear clarification and justification. Sometime commas was used instead the dots in table 5 and 6. The conclusion section should be concise and to the point. Conclusions that are too long often have unnecessary detail. The conclusion section is not the place for details about methodology.
  2. Some specific comments:

Abstract – Needs to be more precise. Show results. Specify the other algorithms.

l 83 – Aforementioned ones?

I 278 - explain Vno and Fmin ?

I 321 – Min-Max 0-0, 3-3?

I 334- No Free lunch theory, reference is needed.

I 378, 379- RR, LYM, etc. State the all words of any abbreviation used.

I-469, I-486, I-509, I-52, etc. references need to be written completely (year!).

Responds:

  1. We edited, firstly we develop the optimization algorithm and try to demonstrate the our algorithm that’s why we need to comparison also after combining the proposed algorithm to the DNN, we should try more demonstrate and apply on the real dataset. The order of the sections is edited and for more understanding the reader we added some more information in sections. And conclusion also edited as small as possible.
  2. Results are added to the abstract.

“Aforementioned ones?” is deleted from the manuscript for clarifying the sentence.

“No free lunch theorem” reference was added.

Vno is the dimension of the function and Fmin is the minimum value of the function.

Min-Max was corrected and function minimum values are added. 

In part 4.3 was modified as below in the text:

“All data are acquired under the direction of doctors in the intensive care unit. In the study, the patient’s gender and height were taken into consideration, along with the RR(respiratory rate), LYM(lymphocyte), LYM100(lymphocyte ratio), WBC(white blood cell) , neu(neutrophil), plt(thrombocyte), CRP(c-creative protein), Procalcitonin, HR(heart rate), spo2(peripheral capillary oxygen saturation), Temp(body temperature), BPSYS(systolic blood pressure), bpDias(diastolic blood pressure), bpMean(mean blood pressure), Apache2_first(disease severity classification), Sofa_first(sequential organ failure assessment), and Apache2_Mort(disease mortality severity classification).”

The references are written completely.

[59] D. H. Wolpert, W.G. Macready, “No free lunch theorems for optimization”, IEEE Transactions on Evolutionary Computation, Vol. 1, No. 1, pp. 67-82, 1997.

Round 2

Reviewer 2 Report

1. Abstract is updated with the MSE values.

2. Introduction has been updated with the algorithm 

3. Topology of the proposed aglorithm is provided in the section 2

4. Optimization part has been updated

5. Results have been updated

Overall, paper has been improved.